# A Mixed-Methods Analysis of Care Arrangements of Older People with Limited Physical Abilities Living Alone in Italy

**DOI:** 10.3390/ijerph182412996

**Published:** 2021-12-09

**Authors:** Maria Gabriella Melchiorre, Sabrina Quattrini, Giovanni Lamura, Marco Socci

**Affiliations:** Centre for Socio-Economic Research on Ageing, IRCCS INRCA—National Institute of Health and Science on Ageing, 60124 Ancona, Italy; s.quattrini@inrca.it (S.Q.); g.lamura@inrca.it (G.L.); m.socci@inrca.it (M.S.)

**Keywords:** ageing in place, older people, living alone, limited physical abilities, daily living activities, care arrangements, family, frequency and proximity of help, Italy, mixed-methods

## Abstract

Older people with limited physical abilities, who live alone without cohabiting family members, need support ageing in place and to perform daily living activities. In this respect, both the available informal and formal care seem crucial. The present study aimed to explore the current role of the care arrangements of older people, especially if they have functional limitations. Qualitative interviews were carried out in 2019 within the “*Inclusive ageing in place*” (*IN-AGE*) research project, involving 120 older people who lived at home, alone, or with a private personal care assistant (PCA) in three Italian regions (Lombardy, Marche, and Calabria). A mixed-methods analysis was conducted. Results showed that support networks are still mainly made up of family members, but also of domestic home help (DHH) and PCAs, friends/neighbours, and public services, albeit the latter provide support in a residual way, while the former is not as intensive as it was in the past. Frequency and geographical/living proximity of help play a role, emerging also as a territorial differentiation. The paucity or absence of support, especially from the family, risks compromising the ability of ageing in place. It seems, thus, necessary to innovate and improve, in particular, home services, also through real formal and informal care integration.

## 1. Introduction

As a result of the population’s rapid ageing, living alone without cohabiting family members in old age has become a crucial topic for policy makers. In fact, it represents a living arrangement from which health, as well as social and relational negative consequences, can arise [1]; in addition, further impacts on the possibility of ageing in place, that is, “elderly people continuing to live at home for as long as possible” exist [2] (p. 2). Research also underlined that ageing in place is a key component of the quality of life of older people [3], enabling them to age integrated in their communities, by avoiding or delaying institutionalization until it becomes strictly necessary [4]. “As people age, they accumulate deficits that are eventually manifested as frailty, disease, or disability” [5] (p. 17). The greatest decline/loss of residual functional/cognitive abilities is observed among older people living alone [6], and this in turn is associated with higher risks of hospitalization [7].

Data from the Italian National Institute of Statistics (ISTAT) indicate that in 2020 (1 January) in Italy the over 65s represent 23% of the total population [8], the highest data at the European level (EU average: 20.6%) [9]. Furthermore, in this country 48% of people living alone are over 65, 32% are in the 45–64 age group, and only 20% are under the age of 45 years [10]. As many as 44% of the over 65s have serious difficulties in the activities of daily living, and of these almost half (47%) live alone [11]. Data from the European Health Survey (EHIS) for 2019 [12] indicate, in particular, a worrying demand for assistance especially of those over 75 years old, with about three million of such individuals (out of about seven million) being seriously compromised in their functional abilities, and presenting restrictions on independent living and related need for support. In this respect, the potential care networks outlining the overall mix of Long-Term Care (LTC), aimed at covering the needs of dependent persons and particularly of older people, are usually proposed in the triple breakdown between formal public, formal private, and informal care [13,14].

Public care intervention mainly concerns in-kind services and financial aids through cash-for-care allowances, provided both at home and in residential facilities [15]. In Italy, the main tool in this respect is represented by monetary transfers/cash allowances, in particular by the National Disability Attendance Allowance (*Indennità di accompagnamento*—IA, EUR 522.10 per month in 2021), available to citizens certified as totally dependent. IA was disbursed in 2018 to about 12% of people aged 65 and over, while home-based in-kind services, which are the Integrated Home Care (health and social) (*Assistenza Domiciliare Integrata*—ADI) and the Home Care Service (social) (*Servizio di Assistenza Domiciliare*—SAD) only accounted for 2.7% and 1%, respectively. About 2% of people aged 65 and older live in residential care facilities [16]. Moreover, the annual per capita expenditure on social interventions and municipal services for this age group was EUR 95 in 2017 [17]. Regarding the public sector, some analysis [18] indicated the existence of different territorial welfare models in this country, which may differently affect the possibility of ageing at home for older people with limited physical abilities: the cash-for-care model in the Centre-South, and the residential care model in the North. It is also to be highlighted that geographical inequalities in overall public home care services are due to the lack of central regulation and inadequate financial support for a local and extensive development [19]. The territorial differentiation that contrasts urban and rural/inner areas should also be considered. These are areas with very different characteristics, with a worse context for the latter, which is more fragile in terms of the composition/provision/accessibility of essential services (e.g., social, health) and has more difficulties in ensuring adequate levels of care and assistance for older people and the disabled overall [20].

Among private home services in Italy, the support of a personal care assistant (PCA) paid for by family is widespread, in particular of the so-called “*badante*”, or, more generally, a Migrant Care Worker (MCW) who has a foreign origin, both living in and on an hourly basis [21,22]. In 2019, data from the Italian National Institute for Social Security (*Istituto Nazionale Previdenza Sociale*—INPS) on regular domestic workers in Italy, elaborated by DOMINA [23], confirm the presence of 407,000 PCAs and 441,000 units for domestic home help (DHH), mainly foreigners (70%, with 41% from Eastern Europe) and women (89%). However, out of the estimated two million domestic workers overall, well over half are irregular, that is, generally working without a formal employment contract. More precisely, PCAs and, especially, MCWs in Italy are considered as workers of the grey market [24], since they are characterised by a weak/precarious employment due to expired residence permits and/or to undeclared labour contracts [25]. It should be stressed that in Italy the IA is frequently used to resort to the private care services market, particularly to hire private PCAs who represent a peculiar “pillar” in our welfare system [26], often used to compensate for an inadequate provision of public support services [27]. It should also be noted that, although, especially, foreign PCAs are often assimilated to private home services, part of the literature keeps them distinct, and in fact defines the LTC as the provision of “informal, formal and unregulated care assistance to older persons by family members, public, private and non-for-profit care services and migrants” [28] (p. 135). Elsewhere in the literature [29] even retains them as informal caregivers.

However, informal carers, as generally defined, are above all family members, i.e., spouses and children, as well as friends and neighbours, who take care of relatives (not only older people) with chronic illnesses, disability, or other long-lasting health, social, or long-term care needs/conditions, in need of in-kind support for basic daily activities [25,30]. Informal caregivers provide unpaid long-term support without a formalized contract, outside any professional or formal setting [31], on the basis of personal motivations and social norms [32]. They have a central role in the organisation of care, even when public or private services are available [33]. Recent studies [34], based on various data sources, e.g., the Survey of Health, Ageing and Retirement in Europe (SHARE) and the European Quality of Life Survey (EQLS), showed that in Europe the proportion of informal caregivers increases with age, with a prevalence of 17% aged 18+ and up to 25.6% aged 50+. The same studies indicated that in Italy informal carers aged 50+ are about 18%. For Italy, it is also estimated that, in particular, family caregivers are at least 7,293,000 (of these, 1,362,000, equal to 18.6%, are older people over 65 years); almost 60% of them are women and aged between 45 and 64 years [16]. According to EPICENTRO [35], in this country more than 90% of help to over 65s in carrying out daily activities indeed comes from family members, and 62% of older people living alone receive support, mainly from their children [36]. Adult children thus provide the most important help and social contacts in old age through physical, financial, and emotional support to their parents [37]. Neighbours and friends assume the role of primary informal caregivers when family members are unable to support an older relative [22], however, acting as an important safety net [38]. In this respect, SHARE data for 2017 [36] indicate, for this country, that 6% of help comes from friends and 8% from neighbours. Moreover, of all people aged 65 and over who receive informal personal care, 73% receive it every day, as well as daily help with both housework (34%) and paperwork (36%). Another piece of informal care is provided under the supervision of formal volunteering associations, whose services (for accompaniment and transport) are sometimes remunerated with symbolic amounts of money or are even provided free of charge, and may represent a considerable element in the overall care mix besides the family as well as the public and private service providers [26].

From a more international perspective, it is to put in evidence that informal care represents about 80% of the total LTC provision in Europe. However, in northern countries there is a more limited attribution of caring responsibilities to family members and a well-established and effective LTC system; whereas in continental ones, intergenerational solidarity is mainly supported by public cash transfers or in-kind services; and in southern ones, the burden of caregiving relies on families and public support (especially concerning in-kind services) is rather lacking [13]. Moreover, in northern European countries, family care is less intensive/frequent [22]; mainly in southern European countries, the scarce public services provision has been compensated by low-cost MCWs, mostly women [33], especially following the “migrant in the family” scheme [24] (p. 272).

It is, furthermore, worth highlighting that the literature, when analyzing social frailty [39], often adds, to its functional dimension, the further relational dimension connected to the absence of help from family members, friends, neighbours, carers, services, and other supports, which are needed to cope with the various daily needs and requirements of older people [1]. The absence of support is indeed crucial for older people with limited physical activities living alone. The caring role of families (for preparing meals, doing housework, and providing transportation, as well as offering emotional support and social relationships) seems, particularly, very important for preventing institutionalization and for enabling older people to remain at home and age in place [40], especially when family members who help them live close by/in geographical proximity, or even in co-residence [13]. The absence of family support is conversely hard for most people in need of care. Childless older people often experience a care gap when becoming dependent/limited and need intense support [41]. Moreover, the lack of friends and neighbours [42], low availability of public care services [43], and scarce utilization of some private home services due to high costs [44], could compromise the opportunity of ageing in place, and could, moreover, make the housing of the older adults living alone as a place of loneliness and neglect [45]. 

In older ages, both individual factors such as living alone at home without cohabitant family members and with difficulties in carrying out the activities of daily life, and contextual factors such as (un)available formal/informal care arrangements and social supports/networks, should thus be considered. Moreover, according to some authors [4], poor health, loss of autonomy, and lack of care, among other factors, are barriers to ageing in place. Starting from this framework, with the purpose of exploring the presence and current role of the family and other care arrangements for older people with limited physical functionalities and living alone in Italy, the paper aimed to answer the following research questions: (1) Which main difficulties do older people encounter in carrying out daily living activities in Italy? (2) What are the characteristics of the main family-based and non-family supports? (3) Are there regional and/or urban/rural differences in the different available support networks? The examination of ageing in place arrangements for older people living alone, with particular reference to opportunities/criticalities in accessing welfare services and other available supports, seems fundamental in a perspective of prevention and management of limited autonomy conditions, inherent in the dynamics of ageing processes, particularly in the light of a high risk of unmet care needs. Indeed, if these issues are not properly addressed, there is a higher risk of compromise older people’s well-being and residual independence. 

## 2. Materials and Methods

### 2.1. Study Design: Area and Participants

The paper proposes some results, which emerged from the qualitative interviews carried out as part of the “*Inclusive Ageing in Place*” (*IN-AGE*) research project. The survey involved a total of 120 older people, interviewed in the Lombardy (North), Marche (Centre), and Calabria (South) Italian regions. Following the “*Three Italies*” scheme [46], these contexts can in fact represent the vertical differentiations (North, Centre, and South), which characterize socio-economic phenomena and levels of development in this country [20]. In this regard, three medium-sized urban areas (by population size) were examined, respectively, in each region, having similar proportions of residents aged 65 and over: Brescia (25%), Ancona (26%), and Reggio Calabria (23%) [8]. Moreover, three inner areas (one in each region), with a total of eight rural municipalities, were explored as a more horizontal reading of territorial differences [20]. Twenty-four qualitative interviews were carried out in each urban context (total seventy-two) and sixteen in each rural site (total 48), as showed in Table 1.

Within the three urban cities and the three inner/rural areas, the most fragile locations were identified using territorial, social, and material vulnerability indicators [47]. The most problematic urban districts, as degraded and poorly served peripheral areas [48], were selected when relevant for at least one of the following dimensions: higher presence of older people aged 75 and over, overall and living alone; share of households living in public housing (*Edilizia Residenziale Pubblica*—ERP); high level of unemployment; and low level of education. The indicators were chosen from the variables made available by the General Census of Population and Housing of 2011 [49], the last available census till the start of the survey. In order to detect the most fragile rural sites, reference was made to the classification proposed within the *National Strategy for Inner Areas* (NSIA), whose methodology, defined by the *Agency for Territorial Cohesion* [50], was developed in order to perimeter areas that are not so easily accessible, characterised by progressive depopulation (e.g., high rate of demographic decline/depopulation in the period 1991–2011), an accentuated ageing process, socio-economic depression, and a poor provision of services [51]. This methodology has made it possible to identify, in particular, the most peripheral and disadvantaged rural municipalities based on the distance from the most served urban poles. 

A criterion based/purposive sampling, and not probabilistic, was used, where the sample units are chosen due to particular characteristics, which enable a deep exploration of the themes of the study [52]. The study sample was built on the basis of the following inclusion criteria: men and women aged 65 years and more, who live alone at home without cohabiting family members, or have a private PCA (living in or with intensive daily/nightly care for at least 28–30 h per week); limited physical functionalities, that is, intermediate mobility between limited/reduced within the home, and outside the home (ability to leave the home at least twice a week with the support of a person or aids); absence of cognitive impairment (ability to answer questions independently); and absence of very close by family members who help (living in the same urban block/rural building). 

### 2.2. Recruitment, Instruments and Data Collection

Persons were recruited through the support of the local sections of Auser (voluntary association for active ageing), professionals/operators of municipal/public home services (SAD), and other local/voluntary associations (Anteas, Caritas). These channels were fundamental in providing useful information to identify older persons who could meet the inclusion criteria in the study (by carrying out initial screening/pre-interview), especially to verify their cognitive capacity and intermediate mobility/degree of autonomy, as indeed deemed by these operators/providers. The various territorial contacts of the channels mentioned above also collected adhesions of potential participants on the basis of a detailed information letter on the study aims, and provided to the research teams the relevant names and contact details of potential respondents (address and telephone numbers). More precisely, these local contacts first checked the inclusion criteria on lists of older people in their possession (e.g., users of SAD and volunteering services), and on the basis of the results of this first verification, a preliminary list of potential eligible persons was created, in which to transcribe summary information (i.e., name/surname, age, address/telephone number). Subsequently, local contacts personally verified the availability of the identified older people to be interviewed, and proceeded to deliver the letter containing all the necessary information for the potential participants regarding the purpose/method of the study, data use, and aspects such as privacy and the anonymity of the information collected. Then, a date for the interview was arranged. In some situations, it was necessary for the interviewers, to be accompanied by the local contacts/representatives, for greater serenity and “psychological safety” of the recruited subjects. In any case, a first verbal consent to be interviewed and “to pass on their own personal details” was obtained, as allowed by the international literature regarding the need for consent to provide personal information to third parties by an organization [52] (p. 96).

A semi-structured interview/topic guide, containing both open-ended and some more structured questions, was used. The first (general) were administered for each of the sections provided, and allowed the interviewees to speak freely and report everything they deemed appropriate, sharing their experiences on the research core issues. The second (more specific), drew/focused attention on particular aspects (if not mentioned by the interviewees) of the topics already explored, to better complete/point out the open answers, sometimes also offering a range of possible alternatives [53]. Both types of questions were provided and included for each of the planned sections of the topic guide (family and housing context, health status and use of services, daily living activities and related limitations, care networks/support for daily living activities, economic situation of respondents, social isolation, and perceived loneliness). Basic socio-demographic aspects were gathered by adding some quick-scan questions. The issues were thus explored with ad hoc questions for the survey, adapted from previous studies [54,55], but also incorporating inputs from other research instruments. In particular, the difficulties in carrying out the activities of daily life were first asked generically (by the question “Do you have difficulties in performing the activities of daily life?”), and then they were better clarified by mentioning to the interviewees the items of both the “*Basic Activities of Daily Living*” (ADLs: getting into/out of bed, sitting/rising from a chair, dressing/undressing, washing hands and face, bathing or showering, and eating/cutting food) and “*Instrumental Activities of Daily Living*” (IADLs: preparing food, shopping, cleaning the house, washing the laundry, taking medication in the right doses and at the right times, managing finances) [56]. We also added two sensory limitations such as difficulty in seeing (does not see enough to recognise a friend in the street/from the other side of the street, or to read a product label, also considering the use of glasses or contact lenses) and hearing (cannot hear enough to listen to a television programme at a volume that does not disturb others, considering the possible use of hearing aids), plus two mobility limitations such as going up/down the stairs without stopping and bending to pick up an object [57,58]. For each activity/function it was also asked to specify whether it was performed in autonomy, with help, or not performed (older people are not able). In this respect the respondents were confirmed as never completely autonomous (they carry out all the activities on their own) or completely dependent (they are unable to do anything autonomously). 

The face-to-face interviews (May–December 2019), lasting 60–90 min, were conducted at home of the participants, by six researchers (five females and one male) with expertise in qualitative data collection (two for each region—one for urban sites and one for rural ones). These selected interviewers, having mainly a background as psychologists, sociologists, and anthropologists, were trained ad hoc on both the objectives and protocol of the study through a methodological training seminar that aimed also to conduct a pilot test by carrying out three test interviews (one for each research team from the respective region), which verified and refined the preliminary thematic framework. Interviews were audio-recorded and transcribed in full/verbatim (from audio to electronic text format) by interviewers, omitting—for privacy reasons—the identity of the respondents (replaced by alphanumeric codes). 

### 2.3. Ethical Considerations

Sampling and administration procedures were performed following the ethical and legal requirements for this type of studies, in particular according to the European General Data Protection Regulation (GDPR) No. 679, of 27 April 2016 [59]. Before starting data collection, the positive opinion of the Ethics Committee, on both protocol and materials of the study, was requested and obtained from Polytechnic of Milan, the coordinator of the research project (POLIMI, Research Service, Educational Innovation Support Services Area, authorization No. 5/2019, 14 March 2019). Moreover, a written informed consent form was signed by participants. It incorporated the contents of the information letter and made an explicit request to authorize the audio recording (in addition to a possible video/photographic recording) of the interview. Strong emphasis was put on voluntariness and confidentiality of participation, and particular attention/reassurance was given to the freedom to withdraw from the study at any time without the need of providing explanations, to answer only questions considered as appropriate, and, again, to privacy/confidentiality and anonymity of the information collected, especially with regard to sensitive data.

### 2.4. Data Analysis

A mixed-methods analysis was performed, as in-depth qualitative analysis of the contents with the addition of simple quantitative elaborations, which are presented in introductory thematic tables for providing an initial synthesis/picture of the results, while always keeping a “qualitative dominant or qualitatively driven mixed-methods” analysis [60] (p. 124). More specifically, the qualitative accounts have first driven the analyses and put in evidence for the main categories to be analysed. Secondly, the quantitative analyses, generally as frequency counts of the main categories, were used not as primary findings with statistical value, but only to summarize and guide the interpretation of recurring patterns of meaning. Even though the qualitative analysis was provided to identify main themes, some ‘outliers’ mentioned only few times were not ignored because they integrate the full understanding of the phenomenon, as further elements forming the whole picture [52]. The analysis of results, mainly descriptive, also includes (when relevant) references to territorial differences (regional/urban-rural), and sometimes the terms North, Centre, and South/Midday are also used to indicate the Lombardy, Marche, and Calabria regions, respectively (the three regions included in the survey).

#### 2.4.1. Qualitative Analysis

For the qualitative analysis, the *Framework Analysis Technique* [61,62] was used, which includes five standard phases: in-depth reading of the transcribed interviews; identification of macro-sub categories/themes; indexing-labelling; construction of the thematic chart with categories (headings and subheadings); and reading and interpretation of the qualitative content [63]. Then, a thematic content analysis of the interviews was carried out [64,65,66,67]. It should be emphasized that these steps are more specific for an inductive (bottom up) qualitative analysis, where the themes are defined starting from the collected data. In our study, a mainly deductive (top down) content analysis was provided, starting from the interview questions, which were based on terms and concepts (categories/themes relevant to the phenomenon to be explored), drawing on the existing international/national literature review and on the experience of researchers [68,69]. Our framework was, therefore, already largely identified and reproduced in the topic guide, with theoretical-based definitions of categories [64], representing indeed the “route map for the journey” [70] (p. 59).

As a preparatory phase for the analysis, a list of main macro- and sub-categories of the topics, and possible labels, was constructed on the basis of the topic guide and the related initial conceptual framework. This “tree” was used to set up the thematic charts (one for each macro-category and related sub-categories, further divided for urban cities and rural sites) by means of Microsoft Excel 2019 sheets, as an overall template for reducing the statements from the interviews [52]. Each chart was a two-way matrix, where rows corresponded to cases/interviewees and columns corresponded to categories. The starting framework was, in any case, refined a little after the preliminary and in-depth/line-by-line reading of the interviews [71]. Thus, in addition to ex-ante key concepts/terms already established (concept driven coding), also further (few) ex-post ones were considered when relevant, and included in the charts, for the purpose of an overall reading of the results (data-driven coding) [72,73]. 

In particular, a manual qualitative analysis was carried out, without the use of any software, as supported by some of the literature [74,75]. Qualitative analysis is indeed a critical thinking process, and although specific software can be useful for managing data, a manual analysis can be of help to become more familiar with the results. Moreover, this approach has been facilitated by the preliminary conceptual framework, as described above [52,76]. In order to provide the overall qualitative analysis, the cell colour-coded process is used that is manageable with Microsoft Excel 2019, by means of a tool to sort the data based on the colour of the cell (the same colour for the same pieces of sentences) [77]. It is, anyway, to clarify that, in order to analyse manually the 120 interviews, we proceeded step by step; first examining the contents for individual sites (three urban cities and eight rural municipalities) and then capturing the results (similar/different) as a whole. Two members from each research team filled in the charts with regard to forty interviews of the respective region (twenty-four urban and sixteen rural), and a merging of the charts from three regions was also provided. Then, researchers of each region/team analysed transversally all 120 interviews, broken down in the charts with regard to some aspects/themes closest to their expertise, and then discussed together the appropriateness of reading/coding the contents. 

For the qualitative analysis, the following categories/themes were examined: daily living activities and related functional limitations; care arrangements/networks for daily living activities (composition, geographical/physical proximity, frequency of help); some characteristics of the PCAs; and economic situation of respondents. As already clarified in the *Introduction* of the paper, in the literature PCAs, often foreign, are considered private home services, but also informal caregivers or even kept as a separate category. In the wake of these divergences, in our analysis we have followed the latter current of thought, mainly because to have or not have PCAs was a selection criterion of the sample, and also to keep them distinct from the more specific DHH. We also do not name PCAs as MCWs because they are not the only foreigners in our study. 

#### 2.4.2. Quantitative Analysis

In order to carry out the quantitative analysis (and besides participants’ socio-demographic characteristics), some qualitative dimensions/categorical variables were further codified in agreement between the research units, thus also performing a quantitative-based qualitative approach [78,79]. Two members from each research team provided the data entry with regard to forty interviews of the respective region, and subsequently a consolidated numeric database (with data from three regions) was built/merged. Then, researchers of each region/team analysed transversally (for 120 cases) the database with regard to some themes closest to their expertise (in addition to socio-demographic dimensions), and discussed together the findings. The quantitative data were summarized using Microsoft Excel 2019, by calculating the related percentages (univariate and bivariate analyses). The qualitative dimensions were simply quantified in terms of absence/presence/frequency of the investigated aspect (e.g., yes/no help from the family, PCA, services, friends/neighbours, daily and weekly help), and also by coding ad hoc some further items (e.g., reasons to hire a PCA). Some more complex quantitative classifications were also used in the analysis. Regarding the aspect of limited physical functionalities (based on 12 ADLs-IADLs, two mobility limitations, plus limitations in hearing and seeing), four grades/levels of limitations were provided (mild, moderate, high, very high) [80]. Moreover, the following were elaborated: share of family help on total help; number of family members living close who help (who live in the same urban city/rural municipality of respondents); and monthly income brackets. To clarify, sometimes the qualitative analysis has been quantified by adding only some absolute values, which are reported along the text (without a reference table), in terms of frequency of labels in certain answers (e.g., for details regarding the PCA) and also using only generic terms such as few, some, and several [81]. Tables show absolute values (*n*) and percentages (%). In some tables the sums of the percentage values do not always correspond to 100 as a result of the rounding of individual figures, and also when the number of responses (numerator) is higher/lower than the number of cases or respondents/interviewees (denominator). 

#### 2.4.3. The Analysis Process Scheme

Table 2 resumes the process of categorization/identification of main themes, codification/labeling, and quantification of data provided in this study, in addition to socio-demographic characteristics of respondents and regional and urban/rural dimensions not mentioned in this scheme. 

#### 2.4.4. Quotations from the Interviews

The overall analysis of results was finally supported/clarified/integrated by meaningful verbatim statements, which emerged in the transcription of the interviews [82,83]. Each quotation was translated and coded/labelled by inserting the first three initials and progressive interview number (1–40) of the respective region (LOM = Lombardy; MAR = Marche; CAL = Calabria). An additional code, regarding the specific rural/urban site, has not been included as potentially identifying information, when in combination with detailed quotations of the respondents. A small amount of editing has been necessary to facilitate the comprehension and to replace names of family members/friends (with level of kinship with the respondent) and cities/regions/foreign nations (with generic terms as other city/region, abroad), in order to respect the confidentiality and anonymity assured to the participants, without, however, altering the meaning of the quotations [84]. Moreover, any not relevant omission is put within round brackets; any word that needs to be added to help comprehension of texts or to replace names of people and locations, is put within square brackets [52]. Finally, abbreviations are not reported within quotations in order to let original words from respondents (apart from EUR = Euro).

## 3. Results

### 3.1. Sample Characteristics

Table 3 shows the main socio-demographic characteristics of the respondents, with regard to urban/rural sites of the investigated regions.

Within the sample, mainly older people over 85 years (especially in Calabria) and only 17 cases under 75 years (mainly in Lombardy), women (more in Lombardy and Marche), with an elementary level of education (particularly in Calabria), and widowed (especially in Calabria and Marche) emerged. Moreover, 14 respondents with a cohabiting PCA and 13 with a PCA for daily and-or/night care were found (on the whole, more present in the South). Respondents showed also a mobility mainly outside the home, although with help (especially in the North). The available monthly income is concentrated within the bracket EUR 600–1500 (similarly in the three regions).

### 3.2. Activities of Daily Living and Physical/Functional Limitations

Older participants often reported difficulties in carrying out the various activities of daily living, both basic (ADLs) and instrumental (IADLs), in addition to mobility and sensory limitations related to seeing and hearing problems. A notable 75% reported at least one activity they are unable to perform, while almost half of the sample was affected by multiple limitations (high/very high physical limitations in three or more functions) (Table 4).

A similar context, on the whole, in both urban and rural sites was found, although the worst situations slightly prevail in the latter, with almost 30% of older people reporting five or more activities they are unable to perform, although another 30% reported slight limitations. The highest number of older people with high/very high limitations (56%) is concentrated in the South, while in the North and in the Centre mainly mild/moderate limitations (53% and 60%, respectively) emerged. 

Older individuals were, thus, not able to provide several activities, and the following were reported as particularly hard (sometimes even by the same respondent): cleaning the house (67 units), shopping (52), and bathing/showering (43). These are indeed either heavier activities, or mainly connected to mobility/agility, which is often quite compromised in our sample. The qualitative interviews carried out for our study underpin these findings with some interesting quotations.

Regarding cleaning the house, the heaviest and most demanding activities (cleaning windows, shutters, upper and lower parts in general) cannot be carried out any more.


*I no longer clean upper things, I no longer climb the ladder.*

*(MAR_18)*



*I no longer go up the stairs to wash the windows.*

*(CAL_18)*



*To clean the house (…) I struggle if things are low.*

*(LOM_15)*


Shopping can be very tiring. When possible, older people make order by phone and ask for home delivery, at least for heavier goods. Otherwise, some respondents can still buy from a street vendor. 


*I struggle to go shopping. My legs hurt (...) I cannot.*

*(MAR_27)*



*It is hard to go shopping (...) To take the bags up.*

*(LOM_29)*



*I [try to] have heavy things brought to me. I cannot do three floors!*

*(CAL_7)*



*A street vendor comes from the village below. He has everything.*

*(MAR_33)*


Even having a bath or a shower at a certain age is particularly problematic, and someone helping with personal hygiene is needed, especially if older persons have a bathtub and are no longer able to enter it. When taking a full shower or bath is not possible anymore, sometimes older people can wash in autonomy only some parts of the body and others are left out.


*To take a bath I have to call someone to get into the bathtub.*

*(CAL_8)*



*I cannot take a shower anymore because I have a bathtub, and now I do not go inside anymore [it’s too high].*

*(MAR_13)*



*I cannot wash my back and feet. Because I cannot bend down.*

*(LOM_17)*


### 3.3. Types of Care Arrangements

When older people have functional limitations, various supports seem available, also of more types for each respondent. The family still plays the main role by providing help to perform daily tasks (78% of cases), especially children (60%), both male and female, with a slight prevalence of the latter (37% vs. 33%). Private services and friends/neighbours (42% for both), public services in approximately one-third of cases, and PCAs (23%) follow. In particular, among private services, support comes mainly from DHH (37%), but also from other occasional forms of private help, e.g., acquaintances, for shopping or bureaucratic matters (to pay for utilities), upon payment of a small amount of money. Moreover, in some cases, for private accompaniment/transportation needs (to have medical visits or diagnostic tests) Auser is of help, upon payment of a little symbolic fee. Among public services, help comes mainly from SAD (23%), and other municipal supports (e.g., family foster care, support administrator, social worker only, meals delivery at home, day care centre, transport service) (Table 5).

Mostly family care is, therefore, provided, especially in rural locations (85% of cases) and in the Centre-South (83% and 85%). Children help more in the South (68%) and in rural sites 65%). There is also greater support available from friends/neighbours in rural areas (48%) and in the Centre (55%). Services (especially public ones) are more present in urban areas and in the Centre-North, while in the South (only 15%) they seem to be somewhat counterbalanced by PCAs (38%), who prevail, moreover, in rural sites (27%). DHH is reported more in the Marche region (45%) and in urban areas (47%).

The qualitative interviews again support, clarify, and integrate these findings with some relevant quotations. 

Concerning children, daughters help when they are free from working commitments, especially for heavy housework or shopping. The sons help, too, even at a distance via the internet, with regard to technical/bureaucratic/financial issues and online shopping.


*There are my daughters, where possible, because they currently work a lot, they come back at 10 pm, they have a hectic life!*

*(CAL_12)*



*If I need to buy anything I ask my daughter to buy it for me.*

*(CAL_14)*



*My daughters come at times, If I have to move something, a big job.*

*(MAR_15)*



*My son (…) is familiar with the internet, he helps me even from afar.*

*(MAR_17)*


We also note a rather gendered care because there is less reliance on sons, than daughters, for caregiving. An older woman even defines herself as being alone, because she has only two sons within her parental support network.


*Sons! (…) Oh my God! When really there was a need one of mine accompanied me for shopping.*

*(LOM_15)*



*I’ve two sons (…) However, since my daughter-in-law died, I was left alone.*

*(LOM_1)*


Some nieces/nephews (most often the children of sisters/brothers), sisters/brothers, cousins, and daughters-in-law have also been indicated as further relatives who help, sometimes as significant points of reference.


*My niece comes if I need anything. I call her on the phone and she comes.*

*(MAR_24)*



*My brother comes if I need to take a shower.*

*(CAL_21)*



*I have two female cousins. One prepares foods and the other deals with cleaning and laundry.*

*(CAL_33)*



*My daughter-in-law always comes to see me. If I’m missing something at home, she brings it to me.*

*(CAL_38)*


Friends often do a lot for respondents, materially and psychologically, and also help them feel less lonely. However, older people also turn to friends in order not to ask for support to their children (especially for minor matters).


*My friend gives me affection, security. She helps me a lot.*

*(CAL_17)*



*My son is busy with his own things (...) I cannot make him come in case I dropped something on the floor. I prefer to ask a friend for silly things.*

*(MAR_3)*


Neighbours are very helpful in everyday life. They sometimes also perform a control/monitoring function. Moreover, often friends and neighbours coincide in later life.


*The neighbour is a good girl, I trust her if I need anything.*

*(LOM_36)*



*If the neighbours do not see I get up in the morning, they call me.*

*(CAL_30)*



*Then there is a friend of mine who lives in my same building. She helps me sometimes (…) We go to the church together on Sunday.*

*(CAL_6)*



*When I need, I call a friend who lives in the apartment above mine.*

*(MAR_8)*


PCAs are all women (except for two cases in Calabria). Fifteen out of twenty-seven are from Eastern Europe, six are Italian, and six are from the Philippines and South America. To hire a PCA is necessary especially in case of widowhood (two units), health problems (seven cases), and falls (three units).


*Then my wife died and I got a personal care assistant.*

*(CAL_8)*



*The personal care assistant has arrived when I had a heart attack.*

*(MAR_40)*



*I’m on a wheelchair because two years ago I fell and broke a femur. Due to this episode, I need a personal care assistant.*

*(MAR_20)*


Moreover, both regular contracts (11 out of 27) and verbal agreements for undeclared work (10 out of 27) emerged, whereas in 6 situations no information was referred in this respect. Especially in rural sites in Calabria, no regular contract was reported.


*We have a verbal agreement with her, there is no contract.*

*(CAL_38)*



*She has no contract, we cannot make it, it’s too [expensive].*

*(CAL_39)*


Domestic workers arrive, especially, when cleaning the house is too hard for respondents.


*I didn’t want to take a domestic worker, but I realize I need her (…) I cannot do certain things, such as cleaning with the ladder, as I used to do.*

*(MAR_13)*



*I have a person who comes for cleaning (...) I cannot do it alone anymore.*

*(CAL_35)*


Concerning SAD, there are several positive accounts. However, there are problems at times if the operators or the service’s organisation change (e.g., different days for the provision of the service), since the older person finds it difficult to adapt.


*The girl from home care service helps me a lot (…) For most of the time.*

*(MAR_10)*



*I have to thank the public assistance, the ladies who clean my house.*

*(MAR_16)*



*If they keep changing them [operators from home care service], I do not want them anymore (...) I’ll manage on my own.*

*(LOM_11)*


Furthermore, volunteering and acquaintances are of some help, even though in few cases.


*I pay five EUR to the volunteers, and they accompany me [to the hospital] to do the blood tests (…) Otherwise I wouldn’t know how to do it!*

*(LOM_34)*



*There is a paid person who takes care of things for me. She also takes me shopping when I need it.*

*(CAL_3)*


### 3.4. Care Arrangements and Income

The older persons interviewed all have a pension (work, survivor, social, or invalidity pension). With regard to the sources and amount of income for the entire sample, and with reference to some supports in particular, i.e., PCA, private DHH, and public SAD, not surprisingly, respondents with a better economic situation show a greater chance of paying for the first two care provision opportunities (Table 6).

In detail, among those who benefit from SAD (in some cases paying a minimum fee to obtain it), 32% have two pensions and only 18% report also the IA. None of them reported annuities.


*I have a work pension (…). The accompanying allowance has not yet been given to me (…) I have social assistance [from public home care service].*

*(CAL_21)*


Those who can pay a housekeeper/domestic worker seem to have a better situation. In fact, in 48% of cases they benefit from two types of pensions. In addition, they can also have the IA and annuities for almost 30% in total (18% and 9%, respectively).


*I have a minimum pension and my husband’s survivor’s pension (...) The accompanying allowance was given to me three years ago (...) I use this money for a person [home worker] who takes also me out for shopping.*

*(CAL_3)*


Those who can hire a PCA more often (63%) report at least a couple of pensions. The IA is also available for almost 40% and annuities for approximately 20%.


*I have a work pension and a survivor’s pension, I also have annuities from a small two-rooms apartment that I have rented (...) In addition to the personal assistant during the day, I also have a man for the night.*

*(CAL_1)*


If we consider economic resources by monthly income brackets (Table 6), we note that only those who have, above all, a PCA, but also DHH, report (in some cases) an income of over EUR 1500 per month. Those who benefit from the SAD remain on lower levels of income. However, the analyses including the income variable are overall affected, in the whole sample, by the general concentration of the respondents in the second income bracket, with few cases exceeding EUR 1500 per month.

It is to further highlight the problem of the cost for the PCA, as referred to by many respondents. It was raised mainly by those who cannot afford this support (44 units).


*You must have a high income to be able to pay a personal assistant.*

*(MAR_31)*



*I have been in need of a personal assistant for some time. Who pays for me?*

*(LOM_26)*



*Not everyone can have a personal assistant, because it implies too high an expense, that cannot be reached.*

*(MAR_4)*


Those who have a PCA only, in 13 cases, mentioned their more or less favorable economic situation in this respect. In seven cases, no problem was reported.


*With my pension I live peacefully, I do not have a rent to pay, I can pay the personal assistant, I have what I need.*

*(MAR_39)*



*With the money I have I can manage all the expenses, both the personal assistant and other.*

*(CAL_10)*


In six situations, respondents state they have instead serious difficulties for paying the salary of the current PCA.


*The personal assistant is expensive, she is draining all my money.*

*(MAR_24)*



*I live on 1300 EUR per month, but 950 are for the personal assistant, so everything else is difficult to pay (…) Unfortunately I have to start making sacrifices.*

*(MAR_34)*



*My pension is low! I have to give up a few things, I only buy the essentials (...) I give the personal assistant 300 EUR a month, I cannot give her more than this.*

*(CAL_8)*


### 3.5. Share of Family Help on the Total and Other Supports

When considering the number of family members who help (e.g., three children) as share of the total aid (of all types) received by the respondents (e.g., three children, one domestic worker, and two friends), we found 26 cases (22%) without family help, 52 cases (43%) with moderate family help (up to 50% of the total), and 42 cases (35%) with strong family help (over 50% of the total). In urban sites, compared to rural ones, the absence and moderate presence of family help prevail, while in the latter a stronger supportive role of the family emerges (Table 7).

A strong family aid emerged above all in Calabria (53%). The greater number of cases without any family support was found in the North (33%), while they are less in the Centre-South (18% and 15%, respectively). 

In some situations, in fact, several family members who help and engaged in care work are reported, e.g., children or nieces/nephews who alternate with different tasks (shopping or cleaning the house).


*There are my daughters and my son (…) Then my nieces also come.*

*(CAL_28)*



*In the family there are mainly my nieces who help me (…) Then my son (…) Another son comes when he can, with my daughter-in-law too.*

*(CAL_38)*


Oppositely, only one case emerged without overall help of any type. She has mild physical limitations, no children, and only a relative living in another province of the region.


*Nobody helps me (…) Anyway I have a strong character (...) I cannot do it today? I’ll do it tomorrow. Currently my Parish is my family, they love me.*

*(LOM_08)*


Concerning the whole sample and the share of family help on the total, in relation to the specific presence/absence of other aids, we note the recourse especially to the PCA, but also to private services (e.g., DHH), mainly when the family is moderately/very present. Differently, especially public services (e.g., SAD), but also friends/neighbours, seem to intervene more often when family help is present only up to half of the total aid, or even when it is totally absent (Table 8).

The words of some respondents confirm and clarify better these situations, where family help is more or less available, despite other types of help. For instance, the coexistence of both more family members and a PCA is highlighted.


*The personal care assistant helps (…) Then one daughter provides shopping and another one follows medical issues.*

*(CAL_24)*



*The personal assistant comes at night (…) A niece comes in the morning and she asks me if I need anything. Then there is another niece who comes in the morning or she calls me to know if I need anything.*

*(CAL_38)*


On the opposite, older persons seem even without family help when SAD or friends are available.


*Now there is the home care operator who cleans and washes the windows and fix the house a bit. Then there is God (...) There is nobody else.*

*(CAL_11)*



*There are only three friendly people who help me (...) With them there is a relationship of faith, we communicate very often (…) Neighbours help me too, I have always found someone of them available.*

*(MAR_35)*


### 3.6. Care Arrangements and Frequency of Help Received

Family members offer daily (or almost daily) help in 36% of cases and the PCA in 23% of cases (living in, daily or almost daily, or as a nightly/almost nightly presence), especially in rural sites, where friends/neighbours are also important to some extent (17%). Daily support from services is more evident in urban areas (33% overall, mainly SAD and DHH). Daily help from children (23% of cases) also prevails in urban sites (26%). Daily help from the family, and especially from children, is also more frequent in the South (58% and 35%), as is the support from both PCAs (38%) and friends/neighbours (23%). On a day-to-day basis, the presence of public services prevails in the North (especially SAD for meals at home or personal hygiene/home cleaning). An intermediate situation has been found in the Centre (Table 9). 

On a weekly level, family support again prevails (51% overall and 37% children), especially in rural areas (69% and 52% children) and slightly more in the South (55%). Help from services also re-emerges especially in urban areas, with public services more prevalent in the North and in the Centre (25% and 23%), while private services (30%) and friends/neighbours (30%) are more present in the latter case. The PCA is always present on a daily basis, and never only weekly (Table 10).

It is worth highlighting that weekly help from private and public services is provided exclusively by DHH and SAD. If we compare the daily and weekly frequency of help, we note that all typologies of support (PCAs excluded) are more present on a weekly basis, in particular family help (51% vs. 36%). At the regional level, children help most on a weekly basis in Calabria and Lombardy (about 40%). 

The words of some respondents clarify how the frequency of help is an issue. It is sometimes sufficient and sometimes not. Some children are present more frequently.


*My youngest daughter comes two to three times a day.*

*(CAL_28)*



*If I need there is always my son who comes almost every day.*

*(CAL_33)*



*He [son] comes often. Every day and sometimes even twice a day.*

*(LOM_1)*


In some cases, children seem to pass only to give a look and check if something is eventually needed.


*My son comes every evening, to see if it’s all okay (…) To see if I’m alive!*

*(MAR_29)*



*My son comes every day, even only for five-ten minutes. He prepares the coffee, asks me if I need anything but then he goes away.*

*(CAL_20)*


In other cases, the family network is not supportive enough, and children in particular seem to show up when they remember, e.g., when sought on the phone.


*My son calls every day, but he comes every 15 days, when he remembers!*

*(CAL_19) *



*My daughter comes when I call her!*

*(MAR_37)*


When sought out, some children even almost seem to be available and arrive promptly in order to help their parents.


*If necessary, even at two in the morning, my son came down.*

*(MAR_4)*



*If I need anything, I tell my daughter to come and she is available.*

*(CAL_9)*


However, it is worthy to put in evidence that older people themselves, in some cases, do not wish to rely too much on their children, to depend on their help. They do not want to disturb them.


*My children help me, even though I always try to avoid bothering them.*

*(CAL_9)*



*I try to do by myself so as not to disturb them [children].*

*(CAL_6)*



*He [son] has his work (…) He cannot always take care of me.*

*(MAR_10)*



*Why should I disturb the children, who work a lot, and have families too, children, wives? This does not suit me!*

*(MAR_22)*


With regard to SAD, respondents sometimes generally complain about the scarce presence/frequency of the referent social worker.


*I have not heard from her for about a month.*

*(LOM_26)*



*She was here only once (...) I do not remember her face.*

*(LOM_15)*


At other times, respondents complain specifically about too few hours of SAD received from service providers, which are considered insufficient.


*I would like more help. It was six hours a week, now it’s gone down to four.*

*(CAL_8)*



*The lady from the cooperative [who is in charge of the home care service] helps me four hours a week. It’s too little. If I wouldn’t manage it by myself, how could I live here? Like a tramp!*

*(CAL_3)*


### 3.7. Geographical/Physical Proximity of Family Members Who Help

With regard only to older people with support from at least one family member (94 units), we see 64% of cases where relatives who help (independently from the frequency, on the whole) are also living close by, that is, live in the same urban city/rural municipality where the older person lives. They are two or more especially in urban sites (45%) and in the Calabria region (56%), whereas no close family members is found mainly in Lombardy and rural sites (67% and 54% respectively) (Table 11).

With regard to the share of family help on total (and without considering territorial subgroups), the related support is strong mainly when two or more members who help live close by the respondents (71%), whereas when no or at least one family supportive relative is close, the help is mainly moderate (68% and 73%) (Table 12).

With regard to the frequency of family help on the whole, and from children in particular (again without considering territorial subgroups), there are two or more close by supporting relatives especially in case of daily help (79% and 56%), whereas no helping family member who lives close by is reported, especially when help is weekly (74% and 65%) (Table 13). 

The words of interviewees clarify, in essence, that when family members/children who help also live close by, they seem more likely to support on a daily basis. Family support seems, thus, facilitated by living proximity (same city/municipality). In the rural areas of the Marche region, some relatives (seven cases) even live very close (even only 200–500 m away) to the older person cared for.


*The most direct help is from my daughter who stays here [lives near to me].*

*(CAL_37)*



*My daughters who live here [close to me] help me.*

*(CAL_28)*



*My daughter-in-law helps me (…) She lives nearby at 300 m.*

*(MAR_31)*


Conversely, when relatives live further away (e.g., another city/municipality of the same province, 22 cases overall) they provide mainly weekly help or less.


*My son on Wednesdays comes up, for shopping and get my medicines.*

*(LOM_38)*



*My son comes here every 10 days.*

*(LOM_28)*


When relatives live in another province, region, or even abroad, family help becomes less frequent, on a monthly basis, or more sporadic (10 cases on the whole). In these cases, especially children are mostly heard on the phone and they come to visit their parents “in person” for the summer/Christmas/Easter holidays, or when there is an urgent need.


*A daughter lives in [another region] (…) She comes once a month.*

*(CAL_37)*



*The daughter who lives [abroad] comes at Christmas and Easter, and in July.*

*(CAL_27) *



*My daughter lives [abroad], but if I have a serious need she arrives.*

*(MAR_18)*


However, it should be considered that children often emigrated for work reasons, especially (but not only) from rural areas of South.


*Youth cannot stay here. There is no life here.*

*(CAL_33)*



*The village is empty. If there was a chance of work, someone would have stayed and, therefore, it would have been more helpful for everyone.*

*(CAL_32)*



*Young people, children, all leave this place to find work.*

*(MAR_25)*


Distance from the family thus seems to weigh heavily, and, without help from close relatives, older people feel sometimes alone.


*I have no close relatives who help me, I am alone (...) I have a domestic worker but I pay for her, it is not the same thing (...) Everyone loves me here but as close relative I have no one here [who lives near to me].*

*(CAL_35)*


Temporary/seasonal geographical proximity/closeness, also with temporary co-residence with children (three cases), seems to offer relief anyway. A closeness, although not definitive, can alleviate some needs and reassure, or it can offer better conditions for some months (e.g., during winter). Such a care arrangement is, in particular, needed to manage situations that are no longer sustainable “from a distance”, due to poor health conditions of older people.


*When it happens [episodes of hypoglycaemia] I need a doctor, I call my daughter (...) This year it went well because I lived near her and I felt safe.*

*(MAR_38)*



*In winter I do not spend much time at my home. Honestly, I go to my daughter’s home [another city in other region], where I already have a warm radiator.*

*(LOM_35)*



*My health is rather bad now (…) For the moment my daughter lives here with me, she accompanies me for medical examinations and medications.*

*(LOM_27)*


## 4. Discussion

### 4.1. Premise: Trustworthiness of the Qualitative Data Analysis

Before starting the discussion of results, it seems important to premise some aspects supporting the trustworthiness of the qualitative data analysis. According to Lincoln and Guba [85], the trustworthiness of the qualitative analysis is based on four fundamental criteria: credibility, transferability, dependability, and confirmability. In our study, the credibility lies in the use of a topic guide based in part on questionnaires already successfully applied in previous studies on older people with need for assistance (e.g., ADLs and IADLs scales, various questions on social/health issues) [54,55]. It was also achieved through frequent peer de-briefing sessions among researchers, all with prolonged engagement/research experience regarding the issue of ageing in place (to define the protocol, the topic guide, and the rules for transcription/data analysis). Dissemination seminars were also organized with various stakeholders and experts in order to compare/validate the preliminary results gradually collected. The transferability of qualitative analysis is to be understood as analytic, since the subjects are selected for their typological, and not statistical, representativeness [86,87]. It was achieved through a careful preliminary literature review as background data [88] and examination of the results of previous studies on the phenomenon, e.g., an ISTAT multi-purpose survey [89]. On this basis, the starting conceptual multidimensional framework was then built [13], and data from the literature were, however, subsequently compared with our results in the discussion of this paper, when relevant. Finally, the dependability and confirmability of the research results, as use of objective replicable methods for the purposes of safety and duration of the results themselves, were obtained through an accurate description of the study protocol (approved by a specific Bioethics Committee), with detailed notes on the data collection and analysis process [88], by justifying the choice of the methods adopted and through the use of transparent procedures. In particular, the use of the cell colour-coded process [77], even with the consolidation of data in subsequent Microsoft Excel worksheets, although a time-consuming procedure, nevertheless prevented the loss of various phases of data analysis (from the most analytical to the most synthetic) and ensured the transparency of each phase performed during the process, displaying the various worksheets. Dependability was further based on the constant collaborative approach and discussion in all steps, with peer and professional interactions also between researchers and interviewers, especially regarding data analysis, to solve collaboratively the few disagreements and reach a whole common/shared vision on the final data set [84,90]. All the aspects mentioned above, in our opinion, represent a relevant premise for an appropriate discussion of major findings of the study, as presented below.

### 4.2. Family Still at the Forefront (But Not like It Used to Be)

The aim of this study was to explore both the presence and the role of the family, as well as of other supports, in the care arrangements/networks available for older people with functional limitations and living alone (without cohabiting family) in Italy. It is to premise that specific data on older people living alone are not always available in the literature; therefore, sometimes general data on those aged 65 and over have also been considered in order to discuss findings. Moreover, although our sample is not probabilistic, we have also reported/discussed some Italian national statistics (e.g., from ISTAT) for a comparison with our findings, when relevant.

Our results showed first of all that most of the older people interviewed (75%) report activities of daily life they are unable to perform, and that a third of them have very high functional limitations (at least five functions are compromised), especially cleaning the house, shopping, and taking a bath or shower. These results are overall in accordance with what is reported by ISTAT [58]. In fact, the greatest difficulties of the Italian older people aged 65 years and over emerge in heavy domestic activities (30%), shopping (17%), carrying out light household activities (15%), and even in taking a bath or shower in autonomy (10%). Other data, again from ISTAT [91], indicate that older people living alone, compared to older people as a whole, express a greater need for help both for personal care (32% vs. 21%) and for domestic activities (53% vs. 37%).

In these situations, the findings of this study highlight that the family still seems to be at the forefront of the care activity, representing the primary and dominant source of help (the basic network), and, especially, this is the case of children (at least one member 78% and 60%, respectively), but also of other relatives (nieces/nephews, sisters/brothers, cousins). Further data [11] specify that, among older persons who live alone, the share of those who receive the help of family members is, in any case, lower (73%) than the older people who live in another type of family context (93%). Some authors pointed out, in particular, that older adults living alone tend to have above all family-restricted or child-based networks, especially in Mediterranean family-oriented countries such as Italy [1]. The literature confirms that in Italy the family remains the strongest provider of care, especially for older people with limited physical functionalities and disabilities, and almost always represents the first choice for obtaining support, especially when compared to the alternatives of scarce public home care and residential services [26]. Very strong intergenerational ties, therefore, persist at the basis of informal caregiving towards older people, with social and cultural norms that establish family responsibilities [92,93,94]. 

In our study, moreover, the living/geographical proximity of family members (same urban city/rural municipality where the older person lives) is of considerable importance in the actual care provided, especially day-to-day, when support is largely provided by close relatives. In the cases where a continuous proximity is not available, care solutions with temporary closeness between the older person and the family member who helps are arranged, depending on the season or in situations of particular urgency, also putting in place strategies to satisfy, albeit temporarily, some care needs otherwise difficult to cover “at a distance”. Some authors highlighted that geographic distance or proximity between parents and their children affects both the nature and frequency of contacts as a key factor for having more help [95] and the life choices of older people, given that it is less likely that they will move elsewhere, e.g., to a care institution/facility, when children live close by them [13,96]. 

Although the narratives of the interviewees still highlight the family as an important pillar of informal caregiving, as guarantor of continuity of care at home, several statements highlight an emerging context that we could define as post-familism, where, for example, some children (who live closer) are more present, while others are more committed/engaged in work (especially if geographically distant), and seem to help when sought, if they can. Moreover, SHARE data for 2017 [36] show that Italy is still a country where the role of the family is crucial, even though the trend in recent years shows a weak but gradual change, with a decrease in the percentage of family caregivers in the face of an increasing demand for care from older people. In fact, the decline in extended families and of co-residence, as well as the greater participation of women in the labour market, have actually reduced the availability/capacity of potential family carers [97]. However, this is a post-familism allowed, to some extent, also by the interviewees themselves, who often do not want to disturb or worry their children in particular, especially if they work and do not have time to take care of them. In this regard some authors pointed out that the support of children has a positive impact on the well-being of older parents; however, if it is excessive, the latter can have feelings of guilt towards the former [98], with a negative consequence on the perceived quality of life and well-being [99]. 

As revealed by our results, the care network indeed extends, going beyond the family, and a complementary help emerges (the integrating network), however important, which comes, in order, above all from friends/neighbours, private services (especially DHH), public services (especially SAD), and PCAs. Data from EPICENTRO [35] indicate that help to Italians aged over 65 with difficulties in performing daily activities, and in addition to the family (94%), comes 21% from PCAs, but also from acquaintances (14%). In addition, less than 3% receive support from public health and social care home workers, and less than 1% receive assistance from a day centre and from voluntary associations. Data from ISTAT [11] show in particular that older people living alone in Italy generally make greater use of paid home aid (44%) and, in particular, of the PCA (31%). 

With reference to the frequency of help, our study also shows greater values for the family, when compared to other types of supports mentioned above, both daily and weekly, thus confirming a function above all complementary of the latter. In fact, it emerges that the recourse, in particular, to the PCA, but also to private support/DHH, is even greater when family members who help, although who are not cohabiting, is moderate/strong (up to 50% and over of the total help received). Our data seem, thus, to suggest that the presence of a strong family network does not make support from a PCA or DHH unnecessary, and, especially, the personal assistant does not substitute but integrates the family, thus generating a crowding-in effect [100]. Basically, a functional division of roles to be performed seems implemented: hardest tasks (personal care/hygiene, house cleaning) are outsourced by families to the PCA, whereas other soft tasks (company, managing finances, transport) remain for the family [101]. However, an opposite situation occurs with regard to friends/neighbours and public services, with our data suggesting that these seem to intervene greatly when family help is present only up to half of the total aid, or even absent altogether. In this case, we cannot speak of a crowding-out effect, in particular between the role of family networks and that of public care services, a circumstance that occurs in some northern European countries where strong public services substitute/relieve the family [100]. However, a sort of “composition” effect between the formal and informal network seems to emerge; this is probably inherent to the eligibility rules for accessing the public services prevailing across the country in this regard, which pre-structure the type of older user profile with assessment of physical limitations but also relational/social and family fragility, and, consequently, generally take care of older people with a very weak overall care network. 

As for public support, which emerged as marginal in our study, especially the SAD, data for 2018 confirm its scarce disbursement at national level, with only 1% of beneficiaries aged 65 and older [16]. Data from ISTAT [17] highlight that in 2017 in Italy the total expenditure for services dedicated to older people aged 65 years and more amounted to approximately EUR 1.3 billion, with 41% of resources for residential structures and a lower 36% for home social services. This, anyways, does not configure a sort of trade-off residential vs. domiciliary care, since in Italy both seem to be lacking, with only 2% of people 65 and older in 2018 living in territorial residential facilities [16]. Furthermore, home care services are unable to satisfy a need for help, which in several cases is also continuous, although they cover the need for qualified assistance [89]. In fact, SAD is often considered to be insufficient to cover the various needs by our interviewees, when delivered for a few hours a week. In this regard, official data available for Italy confirm its low intensity, with a national average of about eight weekly hours of SAD per user [102]. This does not allow for the complete care of older persons with limited physical abilities, and conversely implies, when possible, the presence of a family member or a PCA to also cover primary needs, which are not sufficiently satisfied (e.g., going to the bathroom or eating) [16]. Sometimes our interviewees report further problems if the organization of the service changes (e.g., different days of delivery) or if the operators change, since older people struggle to adapt. Some authors also stress the importance of the presence of service operators that represent a lasting reference point for them [103]. 

With regard to PCAs, (both in house or on hourly basis, and in any case with daily and/or nocturnal assiduity), it is worth pointing out their arrival mainly in case of widowhood, health problems, and falls of respondents. According to ISTAT [104], 6% of Italian families with older people have the support of a private PCA, and this percentage increases to 28% when the older person has serious reductions in personal autonomy and exceeds 40% if the older person is alone, when cohabitation of the private assisting person becomes more frequent. Of note is the use, in particular, of foreign personal assistants as a private solution to manage the growing challenge of caring for older people, which has effectively bridged the gap between the modest public service provision and the reduced long-term care capacity of households [33,105]. In our study the problem of the high cost of this aid also emerged, which is not always easily sustainable, in addition to the presence of irregular work agreements. Various authors indicate that PCAs are often expensive and thus inaccessible for many older people. As a consequence, older people cannot always pay for this support, especially if (family) savings is not available and the socio-economic condition is poor/low, thus exposing them to the risk of not purchasing all of the care they need [13,44]. Furthermore, according to INPS data for 2019 [23], out of a total of about two million domestic workers in Italy, of which over 900,000 are PCAs and more than one million are DHH, the irregular component is high and estimated at around 60%, as already highlighted in the introduction of the paper. Another aspect highlighted by our results is the provision of the IA, especially among those supported by the PCA (compared to users of DHH and beneficiaries of SAD). Institutional data [16] highlight that the great diffusion of economic benefits in Italy and, especially, the IA, has favored the recourse to the private market of assistance, thus increasing the number of Italian families employing PCAs, a circumstance widely documented further in the literature [13,22,25]. On the whole, however, our findings highlight, not surprisingly, that those who can hire a PCA, followed by those who can pay a domestic worker, seem to have a better general economic situation, in terms of more cases of two types of pensions: IA and annuities. Several authors and data argue that overall access to private care services, which are purchased on the market (e.g., DHH and PCA), is generally more widespread among individuals with the highest income, especially older people [11,106,107]. For the most economically disadvantaged older persons, care is, therefore, still strongly centred on family caregiving, and access to the market is reduced, while for those in the upper economic class, a more frequent access to the support from a PCA significantly is of help in reducing the burden on family caregiving [100].

Also, a solid friendship/neighbourhood network, with relationships built over time, seems to be of great help to our interviewees. Some respondents even report that they prefer “to disturb” them than their children, especially for minor matters. It is indeed to highlight that neighbourly relationships are intensified in old age as the need for help in everyday life grows, and the proximity of the home, in particular, facilitates and intensifies these relationships [91]. Informal support based on friendship or neighbourhood ties is almost a capital to spend, especially in a country such as Italy, with a welfare that is heavily based on informal care. Mair [108] indicates that friends have great potential to represent an important source of support for older adults, and, especially, for those without family support report having more friends in their network. Nelson [109], in particular, highlights the value of friendship with the concepts of “friends as family”. However, it is a network that mainly provides psychological and moral support and less material/financial aid [110].

Finally, it should be noted that, probably when help from the public services or the family is not available or not sufficient, and the PCA or DHH is too expensive, older people interviewed organize themselves as they can, sometimes also resorting to occasional support of those volunteering, whose associates accompany them for instance to do medical visits, or to simple acquaintances, who are also foreigners in some cases, for small private paid aid/services, e.g., for various bureaucratic matters, paying bills, or even for shopping. This is also in consideration of the fact that all main typologies of supports (PCAs excluded) are more present on a weekly rather than a daily basis, in particular family help. In this respect, it is interesting to note that, according to ISTAT [91], support networks can generally be represented as a set of concentric circles, with the closest family members at the centre (affective networks), and around first are other relatives and then friends and neighbours on whom one can rely (elected networks). Anyway, it is to be considered that the care networks are also made by people who, in the opinion of the individuals cared for, can intervene in case of need, and thus they can also exclude family figures and otherwise include private services and voluntary associations. 

### 4.3. North vs. South, Urban vs. Rural

The regional territorial deepening of our results shows overall a more marked North-South gradient/contrast in Italy, with greater welfare disadvantage for the latter, and with the Centre often showing an intermediate context, sometimes more in line with one or the other part of Italy, depending on the issue. For instance, the greatest number of older people with high/very high functional limitations was found in the South, while in the North and in the Centre mainly mild/moderate limitations were reported. Institutional data [58] support our findings, with serious difficulties in carrying out personal care activities being recorded in the South for 14% of older people over 65 years, against about 9% in the North. There are also serious difficulties in domestic activities for 39% in the South and only about 25% in the North. With specific regard to the three regions included in our study, previous data [111] highlighted a worse situation in Calabria, where difficulties in domestic activities prevailed for 43% of over 65s, compared to 31% in the Marche region and 25% in the Lombardy region. The highest number of older people with high/very high limitations in the South should, however, also be linked to the fact that the great part of respondents aged 85 years and older was found there. 

Our results also highlight a significant regional inequality in the response to the needs of older people living alone, with family support prevailing especially in the Centre-South (over 80%). Children help more in the South (68%), where even a stronger share of family aid (greater than 50% of total aid) is detected. Conversely, the public services seem to support above all in the North and in the Centre, while it is slightly present in the South (15%), where, however, more PCAs emerged (15 out of 27 were recruited in Calabria). The lower availability and/or need for family members for the care of older people in the North can be linked to the greater female employment in the labour market and consequent less time to devote to caregiving, in the face of greater economic availability for alternative solutions and greater offer of services. In the South, on the other hand, families seem to emerge more, probably also as a consequence of the scarce availability of public care services, in addition to fewer employment opportunities, due to a labor market and to an economic system which are not very dynamic (albeit with exceptions). Other authors confirm that in Italy there is a strong regional differentiation, especially in the availability of public services for the care of older people, which penalizes the Midday, with greater involvement of families in this part of the country [100,112].

With particular regard to public services, data for 2017 [113] indicate a percentage of people aged 65 and more users of SAD (on the resident population aged 65 and over) that is particularly absent in the Calabria and Marche regions (0.6% and 0.5%), with a slightly but higher value in Lombardy (1.2%). Further data for 2017 [17] report a per capita expenditure of the municipalities, for welfare services dedicated to those aged 65 and over, that drops drastically from North to South, and specifically with EUR 82 in Lombardy, 46 in Marche, and 17 in Calabria. In the South there is, instead, a greater coverage of the IA, with a share of beneficiaries rising from about 10% in Lombardy to 13% in the Marche and up to 17.6% in Calabria [16]. This context is also due to both greater demographic and epidemiological pressure overall in the Midday, and to the failure to apply a standardized process for assessing needs at national level [100]. In particular, the three regions included in our study seem to belong (among others) to three different welfare clusters [18]. The cash-for-care cluster, within the Calabria region, where there is a high rate of IA beneficiaries, while home and residential services reveal rates below the national average. The mixed cash-for-care cluster, within the Marche region, where there is still a prevalence of the IA and possible addition of home care services. The residential care cluster, within the Lombardy region, where the availability of beds in residences prevails, while home care services and IA record low rates of older users. It should be pointed out that, according to Martinelli [112], the deep diversity of regional welfare systems is also due to the lower organisational capacity of some southern regions in the integrated delivery of social and health care services. Moreover, a greater availability of IA in the South can in turn incentivize the hiring of PCAs in this part of Italy [114]. 

As for the presence of PCAs, which we actually surveyed especially in the South, results elaborated on INPS data at the end of 2019 [23] show a different concentration of regular figures mainly in the regions of the Centre-North, with 79% (every 100 users 79+) in Lombardy, 11% in Marche, and 4.5% in Calabria. According to the authors, this lower presence overall in the South is partly explained by a lower need, due to the lower female employment and the greater presence of inactive women, thus potentially available for caregiving, in these regions. However, the greater presence in the North-Centre could in part also be attributed to the geographical proximity of the Central-Northern regions to Eastern Europe, that is, the main area of origin of the PCAs. Other authors, without distinguishing between regular and irregular figures [89], report a situation closer to our findings, with 7% of older people aged 75 and more hiring this support in the North, 9% in the Centre, and 10% in Midday. Probably in the South the share of irregular is higher. In the South, however, as already highlighted, there is also the largest number of older persons with serious functional limitations and over 85 years of age, and, therefore, with a greater need for help from these specific assistants, especially if we consider that our results highlight how daily, but also slightly weekly, help from SAD is very low in this part of Italy. According to ISTAT [91], the North-South gradient is even more evident if the levels of expenditure are combined with the variety of public services available, with the first part of Italy offering high and diversified assistance, and the second one is poor in assistance with low levels for both dimensions. 

In the South it should also be noted that, although the role of the family is central, with our results indicating that daily help from family and children and the presence of two or more relatives living close by who help prevail in Calabria region, such a care arrangement is, however, at the limit and not sustainable in the future, also due to the need for many family members (and above all children) to move/emigrate to other contexts to work and survive, thus resizing their care function. Indeed, ISTAT [115] reports that in the last ten years about 483,000 young people aged 20–34 years have moved through Italy along the Midday vs. Centre-North trajectory, against 174,000 who have traveled the opposite route. 

Also, the urban-rural territorial comparison shows a gradient, with a general welfare disadvantage in the second context, where older people with the worst level of functional limitations, albeit slightly, was found. The previous literature supports these findings, by highlighting that rural older adults experience both greater unmet ADLs and IADLs needs as well as greater levels of chronic disease compared to urban older people [116,117]. Moreover, we found the caring family especially in rural sites (85% of cases, children 65%), and with daily and weekly frequency higher than in urban sites. For the rest, help comes from private services (e.g., DHH) and public ones (e.g., SAD), especially in urban sites, and from friends/neighbours and PCAs, especially in rural sites. 

Various authors refer that rural inner areas suffer from geographical, economic, and services marginality, while in urban contexts there is a wider provision of public and proximity services [118,119,120]. Inner areas are in fact peripheral and ultra-peripheral sites of the country, hard to reach zones that are characterized by intense processes of depopulation, socio-economic and housing contraction/depression. They are areas far from the development poles and lacking in the supply of essential services (health, education, and mobility), with less diffusion and frequency of public transport and consequently greater difficulty in reaching health facilities when present, also due to the scarce use of modern communication technologies, especially by older people [20].

Rural older people depend, thus, most on informal care networks such as family, and on traditions of intergenerational care. Networks of families seem the main carers [121], and are fundamental for ageing in place because they intervene where welfare systems are fragmented and/or lacking [2,122]. Anyway, in some cases older adults lack adult children close by as potential caregivers because they live farther away. In fact, due to the lack of infrastructures in rural areas, providing essential services such as healthcare, education, and mobility, the young population/children were more easily forced into exit/emigration dynamics and, therefore, to live in other areas [121,123]. As a consequence of this, and, moreover, to compensate the paucity of services, in rural contexts friends and neighbours also play a crucial role as informal caregivers [117,124]. 

In rural-peripheral zones there is indeed a greater presence of community and proximity relationships [20], and the friendship/neighbourhood ties are important in supporting older people in carrying out daily life activities and for ageing in place. Moreover, according to ISTAT [91] the most traditional type of link indeed remains strong in rural areas, that is, the close parental one, but the support network represented by the neighbourhood also characterises the inner South in particular. For the same family migratory reasons and for the lack of services already highlighted, we can further suppose the greater presence of PCAs detected by our survey in rural sites. In this regard, De Rossi [20] highlights how, among the new inhabitants of rural areas, especially mountain ones, there are many individuals from abroad (e.g., Romania), often following family reunification, but also due to the lower cost of living and major job opportunities, especially in the tertiary sector, and, therefore, also as the PCA. 

### 4.4. Limitations

For a more correct interpretation of results, some methodological and analytical limitations should be pointed out. 

Regarding methodology, the target age of 75 years and over, initially chosen, was more suitable for the themes explored, but various recruitment difficulties required an extension up to 65 years. A cognitive test was not used as preliminary screening, but the assessment of the related status was based only on the information given in this regard by the recruitment channels, then confirmed by the interviewers and by the reference families/caregivers of the older persons interviewed. The territorial samples of respondents are not homogeneous for the following reasons. Regarding the Calabria region, fewer ERP urban areas were included (many buildings have been redeemed since the 2011 census), and the subjects recruited are older and with higher functional limitations. Moreover, probably as a consequence of the latter question, more PCAs were included (compared to the other two regions), who are indeed usually hired to assist the most difficult cases. In the Marche region, the choice of inner areas was conditioned by the earthquake that occurred between August 2016 and January 2017 in central Italy, with hard consequences for many rural sites, which were thus excluded from the study in order not to compromise the comparability with the rural areas explored in the other two regions. In addition, a higher number of cases supported by the public service was found in the Marche region, a situation that is perhaps also due to having recruited several eligible subjects for the study through the SAD, especially in urban areas (that in turn provided contacts with cases managed by other municipal services such as support administrator, meals at home, day centre, and transport service), and in particular through the list (that is used for the social worker’s periodic monitoring, even though without SAD provision) of older people living alone in rural areas. Finally, the research units have also tried to identify the rural municipalities most suitable for a qualitative study in practical terms of effective accessibility/reachability of locations (often in isolated and sometimes inaccessible areas), excluding, for example, sites with travel times by car, from the provincial capital, over 70 min. 

Regarding analysis, the income classes used are wide, and although the interviewees were asked for the monthly punctual income, this was not always reported precisely, preventing a subsequent and more precise/effective modulation of the income brackets themselves. However, the fact that in our sample there is a concentration of cases in the EUR 600–1500 range still seems to reflect the reality enough, since according to ISTAT [125] older people who live alone in 50% of cases do not exceed the threshold of EUR 15,392 per year (EUR 1282 per month). Moreover, despite the limitations inherent in samples that fall below 100 units (72 for urban sites, and 48 for rural ones), the tables show also percentages (%), besides absolute values (*n*), for the sole purpose of making it easier understanding and comparing different territorial contexts, although the latter are sometimes very low, with consequent limitations and cautions in interpreting the former, due to possible sampling biases.

In the light of these limitations, both quantitative and qualitative analyses provided insights into themes that were explored and could serve as bases for future investigations on the topic. In this respect, the trustworthiness, in particular, of the qualitative analysis, which we tried to assure by means of constant collaborative approach and discussion in all steps among research teams involved in the study, could represent a strength. 

## 5. Implications

The social protection network for older people with limited functional abilities and living alone, which emerged in Italy from the *IN-AGE* study, still mainly consists of family members (especially children), the so-called primary network, being the dominant source of help, as an expression of the strong intergenerational ties that still exist in this country, especially in the South, although with a decreasing involvement, mainly due to work commitments and related difficulties in reconciling paid work and caring duties. In fact, we are entering a post-family society, in which families are fragmented, decomposed, and recomposed, all of this leading to a “family warming” in reference to caregiving [126]. The unavailability or the growing scarcity of “family care in place” thus leads to the complementary help from DHH and PCAs (“*badanti*”), friends and neighbours, volunteers and acquaintances, and public services. Thus, patchwork forms of support are emerging, which try to integrate, although not always successfully. PCA in particular is not affordable for everyone, and SAD is increasingly marginal. In spite of these difficulties, ageing in place with the family seems to remain the best solution, making it possible to avoid institutionalization. It is, therefore, necessary to plan adequate interventions, in order to compensate/integrate the decreasing care role of family networks. 

As for specific actions that could be implemented, those for the development of innovative services at home, those for the facilitation of innovation, and overall system actions could work [127]. One route/starting point seems, thus, the strengthening of home care, through a better partnership between formal and informal care services in the LTC field, with the enhancement of community/proximity networks, in addition to the integration of PCAs into the formal care system. There is also a need for actions to facilitate innovation via new technologies (e.g., eHealth), especially for older persons with multimorbidity and living in hard-to-reach rural sites [128]. Furthermore, systemic actions should be envisaged, that is, regulatory, legislative, and financing interventions, to redefine the governance of the following: care integration across health and social care; measures to strengthen the reconciliation between paid work and care; an incentive system (e.g., care allowances) for hiring PCAs with regular contracts [100,129]; and the promotion of active ageing and healthy lifestyles [130]. Moreover, some European approaches recognizing family caregivers as formal co-workers or care providers, and ensuring them dedicated remuneration forms and training opportunities, could also be implemented in Italy [131]. 

The actions mentioned above, however, cannot work without taking into account and tackling, with appropriate policy measures, the territorial differences in Italy that emerged from the study, which are to the detriment particularly of the South and inner areas, especially with regard to the poor support obtained from public home care services (social health care). This context is due to the systematic public disinvestment of the Italian government in the Midday [112], and also to the fragmentation of public LTC-related services between the state, the regions, and the municipalities [132].

Finally, more research seems needed on the phenomenon of current care arrangements of older people with functional limitations living alone, for example, with a focus on those who live with their (old) partner, and also with regard to further Italian territorial paradigms, e.g., metropolitan cities. More research is in general fundamental in order to provide caring solutions, which are able to guarantee a true protection in old age, to implement a ‘zero kilometers welfare’, and to obtain the so far non-recognized universal right to be cared. 

## 6. Conclusions

Older people with limited functional abilities face many difficulties in providing daily activities, especially when living alone. Family members still assist them, but the increasing paucity or absence of this support, in addition to insufficient provision of public services, represent a serious risk for ageing in place. It seems, thus, necessary to boost up, in particular, home services, and to integrate formal and informal care, also taking into account the territorial differences, which in Italy highlight a more problematic context in the South and inner areas.

## Figures and Tables

**Table 1 ijerph-18-12996-t001:** Regions and sites.

Regions	Urban Cities	N ^1^	Inner Area/*Rural Municipalities*	N ^1^	Total
Lombardy (North Italy)	Brescia	24	Oltrepò Pavese:	16	40
			*Menconico*	2	
			*S. Margherita di Staffora*	4	
			*Varzi*	10	
Marche (Centre Italy)	Ancona	24	Appennino Basso Pesarese e Anconetano:	16	40
			*Apecchio*	3	
			*Cagli*	7	
			*Piobbico*	6	
Calabria (South Italy)	Reggio Calabria	24	Area Grecanica:	16	40
			*Roccaforte del Greco*	8	
			*San Lorenzo*	8	
Total		72		48	120

^1^ N = number of interviews.

**Table 2 ijerph-18-12996-t002:** The process of categorization, coding, and quantification.

Macro-Categories	Sub-Categories	Codes/Labels for the Analysis	Quantitative Items (N = Number)
Daily living activities	Physical/Functional limitations: Basic Activities of Daily Living (ADLs); Instrumental Activities of Daily Living (IADLs) Mobility limitations: Going up/down the stairs and bending to pick up an objectSensory limitations: Hearing and seeing	Activities performed in autonomy, with help, and not performed (respondents are not able) Cleaning the house, shopping, and bathing/showering	N. of activities that each respondent is not able to perform *Levels of functional limitations:* Mild = no activities “not able” Moderate = one-two “not able” High = three-four “not able” Very high = five or more “not able” N. of respondents not able of cleaning the house, shopping, and bathing/showering
Care arrangements for daily living activities	Type/Composition	Family (e.g., sons/daughters); public services (e.g., home care-SAD); private services (e.g., domestic home help-DHH); private personal care assistant (PCA); friends; neighbours; volunteering; acquaintances	Main types of help for each respondent *Share of family help on total:* No family help Moderate family help (up to 50%) Strong family help (over 50%)
	Frequency of help	Daily, weekly	Main types of help and frequency for each respondent
	Geographical/physical proximity of family members who help	Same urban city/rural municipality where the older person lives; farther temporary proximity	*N. of family members who help living close to each respondent:* none, one, two or more
	PCA	Reasons to hire PCA: widowhood, health problems, and falls of respondents	N. of respondents reporting a reason
		Characteristics of PCA: gender, country of origin, type of employment, and type of contract	N. of females/males N. from East Europe/other country N. of cohabitants/living in/in house N. of not cohabitants/on an hourly basis N. of regular/irregular contracts
Economic situation	Sources of income	Pension, Disability Attendance Allowance (IA), annuities	N. of respondents with two pensions N. of respondents with IA N. of respondents with annuities
	Amount of income	Monthly income	*Monthly income brackets (EUR):* up to 600 601–1500 1501–2500 Over 2500
	Financial difficulties	To pay a PCA	N. of respondents with PCA N. of respondents without PCA

**Table 3 ijerph-18-12996-t003:** Sample Characteristics (absolute values/*n*).

Characteristics	Regions and Sites						
	Lombardy		Marche		Calabria		Total
	Urban	Rural	Urban	Rural	Urban	Rural	
**Age Range**	**68**–**96**	**70**–**90**	**70**–**101**	**70**–**93**	**67**–**100**	**76**–**95**	**67**–**101**
**Age Groups (*years*)**							
67–74	5	4	3	1	4	-	17
75–79	4	3	4	2	2	4	19
80–84	6	4	5	6	4	3	28
85 and over	9	5	12	7	14	9	56
**Gender**							
Male	5	4	5	3	8	5	30
Female	19	12	19	13	16	11	90
**Education**							
No title	1	-	2	7	1	3	14
Primary school (5 years)	10	6	10	5	12	12	55
Middle school (3 years)	5	3	7	2	2	1	20
High school (3–5 years)	8	7	5	2	6	-	28
University/similar (3–5 years)	-	-	-	-	3	-	3
**Marital Status**							
Single	5	2	2	3	4	-	16
Married but not cohabiting	-	1	1	-	-	-	2
Divorced/separated	7	2	2	-	2	1	14
Widowed	12	11	19	13	18	15	88
**Living Situation**							
Alone	23	13	21	11	14	11	93
Cohabitant pers. care assistant (PCA)	1	3	1	2	3	4	14
Not cohabitant/hourly PCA ^1^	-	-	2	3	7	1	13
**Mobility**							
Only in the home	7	5	12	5	10	9	48
Also outside the home with help ^2^	17	11	12	11	14	7	72
**Monthly Income Brackets (EUR)**							
Up to 600	4	1	2	1	1	1	10
601–1500	17	13	16	14	16	13	89
1501–2500	2	2	5	1	5	2	17
Over 2500	-	-	-	-	2	-	2
Missing	1	-	1	-	-	-	2
**Total Cases/Respondents**	24	16	24	16	24	16	120

^1^ Daily/nightly regular attendance for at least 28–30 h a week; ^2^ respondent is able to leave the house at least two times a week, only if accompanied or with aids (cane, walker).

**Table 4 ijerph-18-12996-t004:** Level of physical/functional limitations, by sites and regions.

Level ^1^	Urban		Rural		Lombardy		Marche		Calabria		Total	
	*n*	%	*n*	%	*n*	%	*n*	%	*n*	%	*n*	%
Mild	16	22	14	29	13	33	12	30	5	13	30	25
Moderate	22	31	11	23	8	20	12	30	13	33	33	28
High	18	25	9	19	10	25	8	20	9	23	27	22
Very high	16	22	14	29	9	23	8	20	13	33	30	25
Total respondents	72	100	48	100	40	100	40	100	40	100	120	100

^1^ The level of physical/functional limitations is based on 12 Basic and Instrumental Activities of Daily Living (ADLs-IADLs), two mobility limitations (going up/down the stairs and bending to pick up an object), plus sensory limitations in hearing and seeing. Mild = no activities “not able”, Moderate = one–two, High = three–four, Very high = five or more.

**Table 5 ijerph-18-12996-t005:** Who helps, by sites and regions (at least one type of help) ^1^.

Types of Help ^2^	Urban		Rural		Lombardy		Marche		Calabria		Total	
	*n*	%	*n*	%	*n*	%	*n*	%	*n*	%	*n*	%
Family	53	74	41	85	27	68	33	83	34	85	94	78
*Children* ^3^	*40*	*56*	*31*	*65*	*21*	*53*	*23*	*58*	*27*	*68*	*71*	*60*
*Daughters*	*28*	*39*	*16*	*33*	*15*	*38*	*13*	*33*	*16*	*40*	*44*	*37*
*Sons*	*22*	*31*	*18*	*38*	*9*	*23*	*13*	*33*	*18*	*45*	*40*	*33*
Friends/neighbours	27	38	23	48	15	38	22	55	13	33	50	42
Private services	37	51	13	27	18	45	18	45	14	35	50	42
*Domestic Home Help (DHH*)	*34*	*47*	*10*	*21*	*15*	*38*	*18*	*45*	*11*	*28*	*44*	*37*
Public services	35	49	8	17	14	35	23	58	6	15	43	36
*Home Care (SAD*)	*26*	*36*	*2*	*4*	*12*	*30*	*11*	*28*	*5*	*13*	*28*	*23*
PCA	14	19	13	27	4	10	8	20	15	38	27	23
Total respondents	72	100	48	100	40	100	40	100	40	100	120	100

^1^ The values in the table do not concern the number of family members, friends, etc., who help, but the number of older persons who reported at least one help of the respective type (one case with family helping = even if with more family members helping); ^2^ more types of help/care arrangements are possible; ^3^ both sons and daughters in some cases.

**Table 6 ijerph-18-12996-t006:** Care networks and income of respondents.

Sources of Income ^1^	PCA		DHH		SAD		Total ^3^	
	*n*	%	*n*	%	*n*	%	*n*	%
(At least) Two pensions	17	63	21	48	9	32	47	47
Disab. Attend. Allow. (IA)	10	37	8	18	5	18	23	24
Annuities ^2^	5	19	4	9	-	-	9	9
Total respondents	27	100	44	100	28	100	99	100
**Monthly Income Brackets (EUR)**	**PCA**		**DHH**		**SAD**		**Total** ** ^4^ **	
Up to 600	1	4	5	11	2	7	8	8
601–1500	17	63	32	73	25	89	74	74
1501–2500	7	26	6	14	-		13	14
Over 2500	2	7	-	-	-		2	2
Missing	-	-	1	2	1	4	2	2
Total respondents	27	100	44	100	28	100	99	100

^1^ More sources of income and more types of help/care arrangements are possible. In two cases with PCA, respondents have both IA and annuities. In a further two cases with IA, respondents have both DHH and SAD; ^2^ annuities = income from family businesses, apartment rentals; ^3^ yotal cases respectively with two pensions, IA, and annuities; ^4^ total cases by monthly income brackets.

**Table 7 ijerph-18-12996-t007:** Share of family help on the total, by sites and regions.

Share of Family Help ^1^	Urban		Rural		Lombardy		Marche		Calabria		Total	
	*n*	%	*n*	%	*n*	%	*n*	%	*n*	%	*n*	%
No family help	19	26	7	14	13	33	7	18	6	15	26	22
Moderate family help	33	46	19	40	17	43	22	55	13	33	52	43
Strong family help	20	28	22	46	10	25	11	28	21	53	42	35
Total respondents	72	100	48	100	40	100	40	100	40	100	120	100

^1^ Share of family help = number of family members who help on the total help (from family, private services, public services, PCAs, friends/neighbours). Moderate family help = up to 50% of the total; strong family help = over 50% of the total.

**Table 8 ijerph-18-12996-t008:** Share of family help on the total and other supports ^1^.

Share of Family Help ^2^	PCA		Private Services		Friends/ Neighbours		Public Services		Total	
	*n*	%	*n*	%	*n*	%	*n*	%	*n*	%
No family help	3	11	11	22	14	28	17	40	26	22
Moderate family help	11	41	29	58	30	60	23	53	52	43
Strong family help	13	48	10	20	6	12	3	7	42	35
Total respondents	27	100	50	100	50	100	43	100	120	100

^1^ More types of other supports are possible. Row sums are thus greater than respective row totals, apart from the label “Strong family help” (row sum is smaller than total due to 10 cases with family help only); ^2^ share of family help = number of family members who help on the total help (from family, private services, public services, PCAs, friends/neighbours). Moderate family help = up to 50% of the total; strong family help = over 50% of the total.

**Table 9 ijerph-18-12996-t009:** Daily help, by sites and regions (at least one daily help by type) ^1^.

Daily Help ^2^	Urban		Rural ^3^		Lombardy		Marche		Calabria		Total	
	*n*	%	*n*	%	*n*	%	*n*	%	*n*	%	*n*	%
Family	23	32	20	42	6	15	14	35	23	58	43	36
*Children*	*19*	*26*	*9*	*19*	*4*	*10*	*10*	*25*	*14*	*35*	*28*	*23*
PCA	14	19	13	27	4	10	8	20	15	38	27	23
Friends/neighbours	8	11	8	17	3	8	4	10	9	23	16	13
Public services	14	19	1	2	7	18	5	13	3	8	15	13
*SAD*	*9*	*13*	*1*	*2*	*6*	*15*	*2*	*5*	*2*	*5*	*10*	*8*
Private services	10	14	4	8	4	10	4	10	6	15	14	12
*DHH*	*10*	*14*	*3*	*6*	*3*	*8*	*4*	*10*	*6*	*15*	*13*	*11*
Total respondents	72	100	48	100	40	100	40	100	40	100	120	100

^1^ The values in the table do not concern the number of family members, friends, etc. who help daily, but the number of older persons who have reported at least one help of the respective type and frequency (one case with family who helps daily = even if more family members help with the same frequency). Moreover, more types of daily help/care arrangements are possible, however, compensated by overall fewer cases with daily help. Column sums are thus smaller than respective column totals, except for Calabria region (column sum is greater than total due to more cases of daily help); ^2^ daily help = even if four–six days a week/more or less every day (even if five nights a week/more or less every night for the PCA); ^3^ only one case of daily SAD in Marche region.

**Table 10 ijerph-18-12996-t010:** Weekly help, by sites and regions (at least one weekly help by type) ^1^.

Weekly Help ^2^	Urban		Rural ^3^		Lombardy		Marche		Calabria		Total	
	*n*	%	*n*	%	*n*	%	*n*	%	*n*	%	*n*	%
Family	28	39	33	69	19	48	20	50	22	55	61	51
*Children*	*19*	*26*	*25*	*52*	*17*	*43*	*11*	*28*	*16*	*40*	*44*	*37*
Friends/neighbours	14	19	10	21	7	18	12	30	5	13	24	20
Private services (all DHH)	17	24	6	13	7	18	12	30	4	10	23	19
Public services (all SAD)	21	29	1	2	10	25	9	23	3	8	22	18
Total respondents	72	100	48	100	40	100	40	100	40	100	120	100

^1^ The values in the table do not concern the number of family members, friends, etc. who help weekly, but the number of older persons who reported at least one help of the respective type and frequency (one case with family helping weekly = even if more family members help with the same frequency). Moreover, more types of weekly help/care arrangements are possible. Column sums are thus greater than respective column totals, except for Calabria region (column sum is smaller than total due to six cases without any type of weekly help); ^2^ weekly help = one-three times a week; ^3^ only one case of weekly SAD in Marche region.

**Table 11 ijerph-18-12996-t011:** Number of family members living close by who help, by sites and regions.

Close by Family Members ^1^	Urban		Rural		Lombardy		Marche		Calabria		Total ^2^	
	*n*	%	*n*	%	*n*	%	*n*	%	*n*	%	*n*	%
None close	12	23	22	54	18	67	9	27	7	21	34	36
One close	17	32	9	22	7	26	11	33	8	24	26	28
Two or more close	24	45	10	24	2	7	13	39	19	56	34	36
Total respondents	53	100	41	100	27	100	33	100	34	100	94	100

^1^ Living close by relatives who help = who live in the same urban city/rural municipality where the older person lives; ^2^ respondents with family members who help overall (94 units).

**Table 12 ijerph-18-12996-t012:** Share of family help on total and number of family members living close by who help ^1^.

Share of Family Help ^2^	None Close		One Close		Two/More Close		Total ^3^	
	*n*	%	*n*	%	*n*	%	*n*	%
Moderate family help	23	68	19	73	10	29	52	55
Strong family help	11	32	7	27	24	71	42	45
Total respondents	34	100	26	100	34	100	94	100

^1^ Living close by relatives who help = who live in the same urban city/rural municipality where the older person lives; ^2^ share of family help = number of family members who help on the total help (from family, private services, public services, PCAs, friends/neighbours). Moderate family help = up to 50% of the total; strong family help = over 50% of the total; ^3^ respondents with family members who help overall (94 units).

**Table 13 ijerph-18-12996-t013:** Frequency of family help and number of family members living close by who help ^1^.

Frequency of Family Help	None Close ^2^		One Close		Two/More Close ^3^		Total ^3^	
	*n*	%	*n*	%	*n*	%	*n*	%
Daily help	5	15	11	42	27	79	43	46
*Children*	*2*	*6*	*7*	*27*	*19*	*56*	*28*	*30*
Weekly help	25	74	15	58	21	62	61	65
*Children*	*22*	*65*	*10*	*38*	*12*	*35*	*44*	*47*
Total respondents	34	100	26	100	34	100	94 ^4^	100

^1^ Living close by relatives who help = who live in the same urban city/rural municipality where the older person lives; ^2^ in four cases the family help is monthly or less frequent (column sum is smaller than respective column total); ^3^ both daily and weekly help from family are possible for some respondents (column sum is greater than respective column total); ^4^ respondents with family members who help overall.

## Data Availability

The quantitative data presented in this study are openly available in Mendeley at https://doi.org/10.17632/3ryrpz224h.2 (accessed on 3 December 2021). Original verbatim transcriptions in the charts are not publicly available due to privacy/ethical restrictions; that is, to their containing information that could compromise the privacy/anonymity of research participants. (e.g., including names of persons, locations, and other potential identifiers of respondents).

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
