# Peer review of "A Mixed-Methods Analysis of Care Arrangements of Older People with Limited Physical Abilities Living Alone in Italy"

_ijerph, 2021, doi:10.3390/ijerph182412996_

Round 1

Reviewer 1 Report

Dear authors,

Thank you for your manuscript ‘Care Arrangements of Frail Older People Living Alone in Italy. 2 A Mixed-Methods Analysis’. Unfortunately, I have concerns about it.

Overall:

The manuscript is more about older people with limitations in activities in daily living, than explicitly on frailty. The use of the term 'frailty' is somewhat misleading, because the concept of frailty is used in the manuscript without specifying in the methods section how it is defined.  Common definitions of frailty according to Fried or Rockwood, outline which factors lead to the definition of frailty. This is a clear shortcoming in the manuscript and leaves the reader somewhat confused. In terms of content, it would be appropriate to use 'limited' instead and to explain in the methods section what leads to a classification. I suggest to focus on ‘limited older people’ and skip ‘frailty’.

It is open as to what the actual focus is: a description of the care arrangements and the challenges that go with them or the difficulties that arise in the Italian system due to the different local circumstances.

Specific:

Literature [5] is a paper on the validation of a screening tool. The publications of Fried or Clegg are more helpful in this context. Please rewrite in the introduction.

Fried LP, Tangen CM, Walston J, Newman AB, Hirsch C, Gottdiener J, Seeman T, Tracy R, Kop WJ, Burke G, McBurnie MA; Cardiovascular Health Study Collaborative Research Group. Frailty in older adults: evidence for a phenotype. J Gerontol A Biol Sci Med Sci. 2001 Mar;56(3):M146-56. doi: 10.1093/gerona/56.3.m146.

Rockwood K, Mitnitski A. Frailty defined by deficit accumulation and geriatric medicine defined by frailty. Clin Geriatr Med. 2011 Feb;27(1):17-26. doi: 10.1016/j.cger.2010.08.008

Line 64: please delete 'only', it contains a valuation that is not appropriate here.

Line 84: What does ‚irregular‘ mean? Not legally in Italy? Working without a contract? Please specify.

118 “older people with 65 years and over receive informal personal care every day in 73% of cases”, Please note this sentence could be misunderstood: of all SHARE respondents receiving care, 73% receive daily care. Please rewrite the sentence.

125: Please introduce the term social frailty, see Bunt S., Steverink N., Olthof J., van der Schans C.P., Hobbelen J.S.M. Social frailty in older adults: A scoping review. Eur. J. Ageing. 2017;14:323–334. doi: 10.1007/s10433-017-0414-7

Table 3: no participants aged 65 or 66, please change denominations of age groups (67-74)

Best regards

Reviewer 2 Report

Thank you for the opportunity to review this paper. This paper is interesting and important topic to understand the current role of care arrangements of older people especially if the latter have functional limitations. I enjoyed reading the paper and I hope you find the following comments helpful.

As a whole, would it be possible to summarize the text throughout the manuscript?

Title: title is clear.

Abstract: abstract is clear.

Keywords: keywords are appropriate.

Introduction:

You use different concepts, such as older people and elderly. Please, select one concept and use it through the manuscript. Introduction focuses pretty much on the Italian situation. Is it possible to add aspect that is more international? Otherwise, the introduction is versatile.

Materials and Methods section is clear and detailed. However, description of recruitment process of study participants is stingy. Please, clarify how recruitment conducted in practice. Who the Experts are? I suggest you add the title “ethical considerations” including permission of Research Ethics Committee, voluntary participation etc. These should base on previous literature.

Description of qualitative analysis is long. Could it be summed up?

Please, move the description of Trustworthiness of the Qualitative Data Analysis to discussion section.

Results, discussion and sections are clear.

Conclusions and Implications. Whether it is possible to describe more closely? What are the main points you want to emphasize and highlight?
